# Acute Cardiovascular and Cardiorespiratory Effects of JWH-018 in Awake and Freely Moving Mice: Mechanism of Action and Possible Antidotal Interventions?

**DOI:** 10.3390/ijms24087515

**Published:** 2023-04-19

**Authors:** Beatrice Marchetti, Sabrine Bilel, Micaela Tirri, Giorgia Corli, Elisa Roda, Carlo Alessandro Locatelli, Elena Cavarretta, Fabio De-Giorgio, Matteo Marti

**Affiliations:** 1Department of Translational Medicine, Section of Legal Medicine and LTTA Center, University of Ferrara, 44121 Ferrara, Italy; beatrice.marchetti@unife.it (B.M.); sabrine.bilel@unife.it (S.B.); micaela.tirri@unife.it (M.T.); giorgia.corli@unife.it (G.C.); 2Laboratory of Clinical & Experimental Toxicology, Pavia Poison Centre, National Toxicology Information Centre, Toxicology Unit, Istituti Clinici Scientifici Maugeri IRCCS Pavia, 27100 Pavia, Italy; elisa.roda@unipv.it (E.R.); carlo.locatelli@icsmaugeri.it (C.A.L.); 3Department of Medical-Surgical Sciences and Biotechnologies, Sapienza University of Rome, 00185 Roma, Italy; elena.cavarretta@uniroma1.it; 4Mediterrranea Cardiocentro, 80122 Napoli, Italy; 5Section of Legal Medicine, Department of Health Care Surveillance and Bioetics, Università Cattolica del Sacro Cuore, 00168 Rome, Italy; fabio.degiorgio@unicatt.it; 6Fondazione Policlinico Universitario A. Gemelli IRCCS, 00168 Rome, Italy; 7Collaborative Center for the Italian National Early Warning System, Department of Anti-Drug Policies, 00186 Rome, Italy

**Keywords:** JWH-018, synthetic cannabinoid, cardiovascular, respiratory, atropine, amiodarone, propranolol, nifedipine

## Abstract

JWH-018 is the most known compound among synthetic cannabinoids (SCs) used for their psychoactive effects. SCs-based products are responsible for several intoxications in humans. Cardiac toxicity is among the main side effects observed in emergency departments: SCs intake induces harmful effects such as hypertension, tachycardia, chest pain, arrhythmias, myocardial infarction, breathing impairment, and dyspnea. This study aims to investigate how cardio-respiratory and vascular JWH-018 (6 mg/kg) responses can be modulated by antidotes already in clinical use. The tested antidotes are amiodarone (5 mg/kg), atropine (5 mg/kg), nifedipine (1 mg/kg), and propranolol (2 mg/kg). The detection of heart rate, breath rate, arterial oxygen saturation (SpO2), and pulse distention are provided by a non-invasive apparatus (Mouse Ox Plus) in awake and freely moving CD-1 male mice. Tachyarrhythmia events are also evaluated. Results show that while all tested antidotes reduce tachycardia and tachyarrhythmic events and improve breathing functions, only atropine completely reverts the heart rate and pulse distension. These data may suggest that cardiorespiratory mechanisms of JWH-018-induced tachyarrhythmia involve sympathetic, cholinergic, and ion channel modulation. Current findings also provide valuable impetus to identify potential antidotal intervention to support physicians in the treatment of intoxicated patients in emergency clinical settings.

## 1. Introduction

The novel psychoactive substances (NPS) phenomenon has taken hold in the market for European and extra-European drugs of abuse [1]. These new synthetic substances, which are mainly traditional drug derivatives, result in a challenge for authorities due to their high potency as well as the difficult identification with screening tests [2,3]. The number of NPS has increased during the last decade, especially in the synthetic cannabinoids (SCs) category [4]. Currently, SCs compounds are among the larger and various groups of NPS monitored by the United Nations Office on Drugs and Crime (UNODC) and European Monitoring Centre for Drugs and Drug Addiction [4,5,6].

Smoking mixtures containing SCs have been sold in the drug market since 2008 when JWH-018 was found in “Spice” and “K2”, the typical brand names of SCs herbal mixtures [6,7]. Digitalization and technological progress have made the products available on the web market as well, where they have become popular among adolescents and young adults due to their low price and easy availability [7,8,9,10].

SCs have been found in different cases of intoxication, which mainly report CNS adverse effects, but cardiovascular (CV) and respiratory system damages can also be worrying [11,12,13]. Patients have shown symptoms such as tachycardia, hypertension, arrhythmias, chest pain, palpitations, respiratory acidosis, and dyspnea after assumptions of various SCs brands, such as Spice or K2, containing JWH-018 or Δ9-THC compounds [14,15,16,17,18,19]. Even bradycardic and hypotensive responses have been reported after SCs assumption, especially among the third-generation SCs [12,14,20]. Preclinical in vivo studies mainly revealed that synthetic cannabinoid-induced effects are bradycardia, bradyarrhythmias with sudden tachyarrhythmias, hypertension, and bradypnea [19], as our recent JWH-018 study also demonstrated [21]. Moreover, previous studies in rodents confirmed these effects after JWH-018 administration [22,23] as well as after administration of other SCs (i.e., CP-55,920 or AKB48) [24,25]. Moreover, smoking K2 products led to myocardial infarction in one 16-year-old boy [26]. Death cases have been reported after JWH-018 assumption, including a cardiac arrest case after SCs abuse [27,28,29]. These reports highlight the significant public health concerns regarding use of these NPS and the severity of life-threatening adverse effects due to SCs, thus underling the urgent need to characterize and develop effective treatment strategies for risks associated with both acute intoxication and chronic use. Actually, treatments of intoxication and withdrawal are still supportive and symptomatic, as no specific antidotes are available.

It is well known that CB1 receptor binding is the main cause of the effects induced by JWH-018 in preclinical models. In fact, numerous previous studies have shown that pretreatment with AM-251 prevented these effects [21,30], thus confirming the involvement of CB1 receptors that are highly expressed in CNS of rodents [31]. Centrally, CB1 receptors can also modulate cardiac and vascular functions involving the medulla and dorsal periaqueductal gray (dPAG) [32,33]. The role of CB2 receptors in the mechanisms underlying the effects induced by SCs in mice has been shown [21]. In fact, CB2 receptor selective antagonist, AM 630, reverted CV and respiratory effects of JWH-018 in a different manner from AM 251 [21]. In line with these findings, preclinical and clinical data state that CB receptors are G protein-coupled receptors involved in cognitive, cardiovascular, and metabolic functions. Notably, CB1 is in abundance in the mammalian brain, where it is responsible for psychoactive effects induced by cannabinoids, while CB2 receptors are mainly expressed in immune cells. It is worth noting that, although to a lesser extent, CB1 receptors are also situated in human peripheral tissues, such as the heart and vasculature [34]. CB2 receptors may also be further involved in CV and respiratory disorders observed in humans [35,36].

Previous studies suggested the involvement of both sympathetic and parasympathetic systems to explain cannabinoid-induced CV effects [32,37]. This is confirmed by both previous SCs studies carried out on rodents [32] and studies conducted on volunteers, which showed that tachyarrhythmias evoked by cannabis smoking were prevented by propranolol administration [37].

As shown in most recent research, modulation of CB1 and CB2 receptors could lead to interaction with ion channels [21,38]. Every phase of the cardiac reaction is regulated by voltage- and ligand-gated ion channels, which can be implicated in many types of arrhythmias due to, for instance, autonomic nervous system overstimulation or electrolyte imbalance [39]. Ion channels can be altered via CB receptor-dependent pathways that, through adenyl-cyclase inhibition, lead to sodium channel, calcium channel, and potassium channel function modulations [40]. These compounds can also bind directly to ion channels, such as Δ9-THC, a cannabinoid receptor partial agonist that inhibits T-type calcium channels, CaV3.1, CaV3.2, and CaV3.3 with pEC50 of 5.81 + 0.02, 5.88 + 0.03, and 5.37 + 0.02, respectively [41]. Delta-9-THC is also able to inhibit voltage-gated sodium channels, as shown in studies carried out on neuroblastoma cells, and more recently, in rat ventricular myocytes [42,43]. Moreover, the interaction between endocannabinoid and potassium channels was reported in an in vitro study, which demonstrated that the endogenous cannabinoid anandamide (AEA) and arachidonoylglycerol (2-AG) blocked cardiac voltage-gated potassium channel (hKv1.5) with IC50 in the micromolar range on mouse fibroblasts (Ltk2 cells) [44].

Cardiovascular adverse effects can be caused by the interaction with CB receptors, but these can also be mediated by interaction with other substrates. Given the previous studies, which indicate the interactions between Δ9-THC and other endocannabinoids with ion-channels or with adrenergic and cholinergic receptors, the purpose of this investigation is to assess how JWH-018-induced CV responses are modulated with drugs that directly act on cardiac substrates. The evaluation has been performed by administering JWH-018 alone and by co-administration with amiodarone (class III antiarrhythmic drug), nifedipine (calcium channel blocker), atropine (anticholinergic drug), and propranolol (beta-blocker), drugs that are and have been widely used in emergency departments. This study also provides a valuable indication to identify potential antidotal intervention to support physicians in the treatment of SCs intoxicated patient in emergency clinical settings.

## 2. Results

### 2.1. Vehicle

To limit the number of mice, the same vehicle was used for all experiments. In vehicle-treated mice, basal HR (663 ± 3.8 bpm; Figure 1A), PD (222 ± 17 µm; Figure 1B), BR (277 ± 7.1 brpm; Figure 1C), and SpO2 (99.1 ± 1.2% SpO2; Figure 1D) did not change compared to the control (untreated) animals over the six-hour observation period.

### 2.2. JWH-018

As previously reported, JWH-018 (6 mg/kg) [21] rapidly reduced HR, inducing deep bradycardia and bradyarrhythmia alternated by sudden episodes of tachyarrhythmia that persisted up to six hours. In addition, PD values significantly decreased compared to basal values immediately after JWH-018 injection and during the last hours of the experiment due to a vasoconstrictor effect, which is also demonstrated by the systolic blood pressure increase. Moreover, systemic administration of JWH-018 rapidly induced a deep bradypnea combined with a transient reduction of the SpO_2_ during the first hour of JWH-018 treatment [21].

### 2.3. Amiodarone

Amiodarone (5 mg/kg) administration by itself slightly induced an oscillatory effect on HR after one and three hours from the injection. The JWH-018-induced long-lasting effect on HR did not revert by amiodarone, which further reduced HR one hour after administration (Figure 1A, significant effect of treatment (F_3,1872_ = 1454, *p* < 0.0001), time (F_71,1872_ = 36.57, *p* < 0.0001), and time × treatment interaction (F_213,1872_ = 9.388, *p* < 0.0001)). Otherwise, amiodarone was able to counteract the insurgence of tachyarrhythmias (Figure 2A–F), especially during the last three hours of the experiment (Figure 2D, significant effect of treatment (F_1,112_ = 39.22, *p* < 0.0001), bin (F_7,112_ = 6.018, *p* < 0.0001), and bin × treatment interaction (F_7,112_ = 4.978, *p* < 0.0001); Figure 2E, significant effect of treatment (F_1,112_ = 80.89, *p* < 0.0001), bin (F_7,112_ = 10.14, *p* < 0.0001), and bin × treatment interaction (F_7,112_ = 9.521, *p* < 0.0001); Figure 2F, significant effect of treatment (F_1,112_ = 86.18, *p* < 0.0001), bin (F_7,112_ = 10.25, *p* < 0.0001), and bin × treatment interaction (F_7,112_ = 9.353, *p* < 0.0001)).

Treatment with amiodarone alone transiently reduced PD during the first hour of the experiment and it slightly reverted the JWH-018-induced vasoconstriction during the fourth hour (Figure 1B, significant effect treatment (F_3,1872_ = 336.6, *p* < 0.0001), time (F_71,1872_ = 3.769, *p* < 0.0001), and time × treatment interaction (F_213,1872_ = 2.654, *p* < 0.0001)). Systemic amiodarone administration induced a slightly oscillatory effect on BR, but it restored the effects caused by JWH-018 during the first and second hours of the experiment (Figure 1C, a significant effect of treatment (F_3,1872_ = 680.6, *p* < 0.0001), time (F_71,1872_ = 3.785, *p* < 0.0001), and time × treatment interaction (F_213,1872_ = 4.496, *p* < 0.0001)). Finally, concerning oxygen saturation, amiodarone did not change the vehicle-treated mice or the transient JWH-018-induced effect in pretreated mice (Figure 1D, effect of treatment (F_3,1872_ = 73.00, *p* < 0.0001), time (F_71,1872_ = 9.094, *p* < 0.0001), and time × treatment interaction (F_213,1872_ = 3.636, *p* < 0.0001)).

### 2.4. Atropine

Administration of atropine 5 mg/kg did not alter basal parameters on HR, but it reverted the bradycardic effect induced by JWH-018 during the last three hours of the experiment (Figure 3A, significant effect of treatment (F_3,1872_ = 747.2, *p* < 0.0001), time (F_71,1872_ = 14.18, *p* < 0.0001), and time × treatment interaction (F_213,1872_ = 6.370, *p* < 0.0001)) and it reduced tachyarrhythmia episodes, in particular during the third, the fifth and the last hour of experiment (Figure 4C, significant effect of treatment (F_1,112_ = 4.483, *p* = 0.0006), bin (F_7,112_ = 12.95, *p* < 0.0001), and bin × treatment interaction (F7,112 = 4.034, *p* = 0.0364); Figure 4E, significant effect of treatment (F_1,112_ = 58.89, *p* < 0.0001), bin (F_7,112_ = 10.74, *p* < 0.0001), and bin × treatment interaction (F_7,112_ = 8.571, *p* < 0.0001); Figure 4F, significant effect of treatment (F_1,112_ = 87.34, *p* < 0.0001), bin (F_7,112_ = 9.646, *p* < 0.0001), and bin × treatment interaction (F_7,112_ = 9.995, *p* < 0.0001)).

Regarding PD (Figure 3B), atropine did not significantly alter the basal values, but it increased pulse distension in pretreated JWH-018 mice during the first and the last two hours of observation (effect of treatment (F_3,1872_ = 128.6, *p* < 0.0001), time (F_71,1872_ = 1.082, *p* = 0.3015), and time × treatment interaction (F_213,1872_ = 1.415, *p* = 0.0002)). On BR (Figure 3C), atropine did not vary basal parameters in vehicle-treated mice, and it partially reverted the bradypnea effect induced by JWH-018 (effect of treatment (F_3,1872_ = 1538, *p* < 0.0001), time (F_71,1872_ = 19.02, *p* < 0.0001), and time × treatment interaction (F_213,1872_ = 9.445, *p* < 0.0001)). Even about oxygen saturation (Figure 3D), atropine by itself did not modify vehicle values and it also did not revert the transitory effect induced by JWH-018 in pretreated mice, but a further decrease of ~10% was seen in the last hour of the experiment (effect of treatment (F_3,1872_ = 78.60, *p* < 0.0001), time (F_3,1872_ = 7.647, *p* < 0.0001), and time × treatment interaction (F_3,1872_ = 4.516, *p* < 0.0001)).

### 2.5. Nifedipine

Nifedipine (1 mg/kg) did not affect the basal HR in vehicle-treated mice, and it did not modify the effect provoked by JWH-018 injection (Figure 5A, effect of treatment (F_3,1872_ = 961.3, *p* < 0.0001), time (F_71,1872_ = 32.85, *p* < 0.0001), and time × treatment interaction (F_213,1872_ = 4.773, *p* < 0.0001)).

The JWH-018-induced tachyarrhythmias were slightly reduced after nifedipine injection in the last two hours of experiment (Figure 6E, effect of treatment (F_1,112_ = 6.417, *p* = 0.0127), bin (F_7,112_ = 13.66, *p* < 0.0001), and bin × treatment interaction (F_7,112_ = 1.945, *p* = 0.0689); Figure 6F, effect of treatment (F_1,112_ = 7.537, *p* = 0.0070), bin (F_7,112_ = 11.41, *p* < 0.0001), and bin × treatment interaction (F_7,112_ = 4.934, *p* < 0.0001)).

Nifedipine administration itself induced an increase of PD (Figure 5B) of ~50% with respect to basal values. Despite this, the administration of nifedipine after JWH-018 injection slightly reverted the vasoconstrictor effect caused by the latter, only during the first 30 min of the experiment (Figure 5B, a significant effect of treatment (F_3,1872_ = 867.6, *p* < 0.0001), time (F_71,1872_ = 3.378, *p* < 0.0001), and time × treatment interaction (F_213,1872_ = 5.162, *p* < 0.0001)). This effect was confirmed by the analysis on BP-2000 that registered the blood pressure changes in one hour. In particular, both systolic (Figure 5E, significant effect of treatment F_3,28_ = 92.29, *p* < 0.0001) and diastolic (Figure 5E, significant effect of treatment F_3,28_ = 92.29, *p* < 0.0001) blood pressure of JWH-018 were reduced by nifedipine administration. Nifedipine slightly reduced basal BR one hour following the treatment, and in JWH-018-pretreated mice, it abolished the reduction of breath rate during the first hour of the experiment (Figure 5C, a significant effect of treatment (F_3,1872_ = 654.9, *p* < 0.0001), time (F_71,1872_ = 8.586, *p* < 0.0001), and time × treatment interaction (F_213,1872_ = 3.623, *p* < 0.0001)). The effect on oxygen saturation subsequent to nifedipine administration did not reveal a difference from the vehicle. However, nifedipine was able to slightly increase oxygen saturation immediately after injection when it was administered after JWH-018 (Figure 5D, the effect of treatment (F_3,1872_ = 4.110, *p* = 0.0064), time (F_71,1872_ = 7.553, *p* < 0.0001), and time × treatment interaction (F_213,1872_ = 3.959, *p* < 0.0001)).

### 2.6. β1 β2 Blocker

Administering propranolol, a non-selective β1 β2 blocker (2 mg/kg), by itself did not significantly change the HR compared to the basal rate. Nevertheless, propranolol systemic administration (2 mg/kg) after JWH-018 injection firstly reduced and subsequently increased the bradycardic effect induced by JWH-018 (Figure 7A, significant effect of treatment (F_3,1872_ = 476.3, *p* < 0.0001), time (F_71,1872_ = 23.55, *p* < 0.0001), and time × treatment interaction (F_213,1872_ = 5.412, *p* < 0.0001)).

Moreover, propranolol drastically reduced the JWH-018-induced tachyarrhythmic events immediately after injection up to the end of the experiment (Figure 8C, significant effect of treatment (F_1,112_ = 36.99, *p* < 0.0001), bin (F_7,112_ = 7.018, *p* < 0.0001), and bin × treatment interaction (F_7,112_ = 6.650, *p* < 0.0001); Figure 8D, significant effect of treatment (F_1,112_ = 33.68, *p* < 0.0001), bin (F_7,112_ = 6.926, *p* < 0.0001), and bin × treatment interaction (F_7,112_ = 3.425, *p* = 0.0023); Figure 8E, significant effect of treatment (F_1,112_ = 20.80, *p* < 0.0001), bin (F_7,112_ = 8.628, *p* < 0.0001), and bin × treatment interaction (F_7,112_ = 4.869, *p* < 0.0001); Figure 8F, significant effect of treatment (F_1,112_ = 40.74, *p* < 0.0001), bin (F_7,112_ = 8.954, *p* < 0.0001), and bin × treatment interaction (F_7,112_ = 5.122, *p* < 0.0001)).

The treatment with propranolol alone slightly decreased the pulse distension, especially after 150 min from the injection, but it did not alter pulse distension reduction induced by JWH-018 (Figure 7B, effect of treatment (F_3,1872_ = 257.5, *p* < 0.0001), time (F_71,1872_ = 2.773, *p* < 0.0001), and time × treatment interaction (F_213,1872_ = 1.738, *p* < 0.0001)). After propranolol alone administration, basal parameters of BR (Figure 7C) were slightly reduced of ~20%. Propranolol was able to increase BR in JWH-018 pretreated mice, immediately after injection, while during last two hours it further reduced the effect caused by JWH-018 (Figure 7C, significant effect of treatment (F_3,1872_ = 826.0, *p* < 0.0001), time (F_71,1872_ = 12.69, *p* < 0.0001), and time × treatment interaction (F_213,1872_ = 4.369, *p* < 0.0001)). Finally, the effect on SpO2 after propranolol administration by itself did not change respect to vehicle-treated mice; however, immediately after propranolol injection, the effects of JWH-018 were abolished (Figure 7D, significant effect of treatment (F_3,1872_ = 21.05, *p* < 0.0001), time (F_71,1872_ = 7.185, *p* < 0.0001), and time × treatment interaction (F_213,1872_ = 4.211, *p* < 0.0001)).

## 3. Discussion

The use of both natural and synthetic cannabinoids has dramatically increased in the past decades due to legalization, to the diffusion as mass culture and, more recently, to the diffuse increase in substance abuse during the COVID-19 pandemic and restrictive measures [45,46]. The recent change in the attitude and the increase in consumption have shed light on the increased CV risk in cannabinoid consumers, as demonstrated by the recently published clinical statements by the American Heart Association and the European Association of Preventive Cardiology [47,48], which highlighted the increased risk of myocardial infarction, arrhythmias, sudden cardiac death, and stroke. Therefore, it is of utmost importance to evaluate the possible pharmacological antidotes, their efficacy, and possible harmful effects.

Currently, no antidote is available for SCs poisoning and the cardiotoxic effects due to NPS. In addition, specific practice guidelines still need to be developed for intoxicated patients. Supportive care and symptom management are the mainstay of the treatment; applying intravenous fluids to treat electrolyte and fluid disturbances is particularly critical to cardiac function. Since patients rarely fit precisely into a particular toxidrome, but rather present overlapping signs and symptoms from manifold syndrome groups, the development of a prompt differential diagnosis is difficult and challenging, requiring relevant cardiology and neurology evaluation. Rapid and consensual treatment of agitation is crucial, also for the healing of cardiovascular toxic effects [3]. In particular, in the pharmacological management of SCs intoxication, the initial treatments of severe hypertension include nitrates, benzodiazepines (BDZs), α-adrenergic antagonists (e.g., prazosin and phentolamine), and β-blockers (e.g., labetalol), these latter used with caution, since potential paradoxical hypertension may occur. Atypical antipsychotics, e.g., haloperidol, ziprasidone, quetiapine, and olanzapine, results are more beneficial than BDZs as fist-line medicaments, even seldom the use of these medications have the potential to worsen SCs-induced QTc prolongation, possibly causing additional cardiac complications, including Torsades de Pointes [49,50].

Moreover, naltrexone, nabilone and naloxone have been proposed lately as potential pharmacotherapy for treating SCs withdrawal [51,52]. Recently, many studies have indicated that prophylactic treatment with CB1 receptor antagonists can block cannabimimetic effects both in animals and humans. Therefore, single-use CB1 receptor inverse agonists could perhaps provide an acceptable temporary single-dose antidote [53,54]. Nonetheless, a never-ending effort is devoted to evaluating the effectiveness of drugs usually employed in Emergency Departments (EDs) for the treatment of NPS-induced adverse effects, with the goal of identifying novel, effective, antidotal therapeutic strategies to be adopted in the critical management of SCs intoxicated patients.

Considering the CV and respiratory effects in mice reported in our preclinical study [21], this work evaluated the interaction between JWH-018 effects and different CV drugs that can be used as symptomatic “antidotes” such as amiodarone (class III anti-arrhythmic drug), nifedipine (calcium channel blocker), atropine (anticholinergic drug), and propranolol (beta-blocker drug), with the aim of counteracting JWH-018-induced effects, by administering cardio-active drugs. Specifically, concerning the choice of currently employed selected antiarrhythmics and antihypertensive medicines, it has to be highlighted that amiodarone is the most commonly prescribed antiarrhythmic drug (AAD) and the most used drug in EDs and Critical Care Units. The other tested AADa, i.e., atropine, nifedipine, and propranolol, are well known to be recommended by clinical practice guidelines for their use in combination with the abovementioned amiodarone or as an alternative effective treatment in the acute management of adverse cardiovascular outcomes (e.g., arrhythmias, refractory ventricular fibrillation/tachycardia) commonly used in EDs [55,56,57] when narrowing an early differential diagnosis enabling the identification of a NPS-induced toxidrome and achieving patient-centered decision making are even more challenging.

Our results showed that only atropine has completely reverted JWH-018-induced bradycardia, and it was the only tested antidote that reverted the JWH-018-induced vasoconstriction. All antidotes reduced the tachyarrhythmia onset induced by JWH-018. All drugs tested, in particular amiodarone administration, reverted respiratory rate. The effect of these substances will be further examined below.

### 3.1. Amiodarone

The first substance examined was amiodarone, which is a class III anti-arrhythmic drug and one of the most used antiarrhythmic drugs in the emergency department [58], acting both on ion channels (K^+^, Na^+^, and Ca^2+^) and adrenergic receptors [59] in the sinus node and in the atrio-ventricular (A-V) node. Amiodarone, administered as a dose of 5 mg/kg, slightly worsened JWH-018-provoked (6 mg/kg) bradycardia, in the central hours of the experiment (Figure 1A). On the contrary, it prevented the onset of tachyarrhythmic events, especially during the last three hours of the experiment, dropping the number of events to zero (Figure 2D–F). The JWH-018-induced effect on PD was not significantly changed during the first two hours after amiodarone administration, but it slightly reverted in the fourth hour of the experiment (Figure 1B). Moreover, amiodarone reverted the reduction of BR caused by JWH-018, but it did not change the JWH-018-induced SpO2 reduction (Figure 1C,D).

In vivo cardiac effects caused by JWH-018 (6 mg/kg), as bradycardia interspersed by sudden tachyarrhythmia events, could be caused by CB1, which can lead to autonomic nervous system modulations and ion channel disbalances [60]. An electrophysiological study showed that amiodarone slows upstroke velocity of action potential and decreases excitability and conductibility of cardiac cells inhibiting K^+^, Na^+^, and Ca^2+^ currents [61]. Beyond these mechanisms, amiodarone is able to diminish sympathetic tone with α and β receptor blockage [56,62]. Action on ion channel and sympathetic tone decrease might be able to further increase JWH-018-induced bradycardia, but also reducing tachyarrhythmic spikes. Although amiodarone seems to worse the bradycardic effect on HR in JWH-018 pretreated mice, this effect could be advantageous in humans. Differently from mice, the most common cardiac symptom reported following SCs use is tachycardia, due to different sympathovagal responses probably caused by the lower dosage taken [16,63]. Indeed, two clinical cases reported that treatment with amiodarone after cannabis or SCs assumption was effective to regularize cardiac rhythm and resolve tachycardia [64,65] due to its direct effect on sinus-atrial node frequency. Likewise, amiodarone regularized all HR changes (arrhythmias and tachycardia) induced by JWH-018 on mice [21]. The inhibition of adrenergic receptors could also explain respiratory response trend following amiodarone administration [66,67,68]. Amiodarone induced tachypnea administered by itself, but also reverted JWH-018-induced bradypnea. This is consistent with clinical cases, which reported tachypnea after amiodarone assumption [69,70].

### 3.2. Atropine

Another attempt to revert the effect of JWH-018 was made by administering 5 mg/kg of atropine, a competitive reversible antagonist of the muscarinic acetylcholine receptors [71]. Atropine was widely used in clinical cases to counteract cardiac abnormalities such as brady-asystolic cardiac arrest [72] or acute myocardial infarction [73] or A-V block [74]. The data obtained seemed to indicate that atropine is able to shorten the duration of the effect of JWH-018 (6 mg/kg) on heart rate (Figure 3A), and also to reduce vasoconstriction (Figure 3B). JWH-018-induced BR reduction was reverted after atropine administration (Figure 3C), while SpO2 did not show significant changes (Figure 3A). Moreover, atropine seemed to decrease the number of tachyarrhythmic events registered after its injection, mainly during the last two hours (Figure 4C,E,F).

The choice of trying to revert JWH-018-induced CV effects with atropine was made in line with the evidence that reported a possible involvement of the vagus nerve, due to cannabinoid receptor activation, resulting in bradycardia [60]. As expected, the increase in HR recorded after atropine treatment confirmed the involvement of vagal activity in the bradycardic effect after administration of JWH-018 (6 mg/kg). Acting as an antagonist for muscarinic receptors, atropine blocks the stimulation of the parasympathetic system with its vagolytic action on sinus-atrial (S-A) and A-V nodes, increasing heart rate [75,76,77,78].

Beyond the expected effect on HR, atropine was also able to decrease JWH-018-induced tachyarrhythmia, particularly during the last two hours of the experiment. By increasing the heart rate, atropine was effective in reducing ventricular ectopic beats, thus improving cardiac dysrhythmia [71,79]. Moreover, atropine decreases the cardiac automaticity in the S-A node [80], improves A-V nodal conduction, avoiding A-V block, which has often been found after SCs administration [81,82]. This evidence indicates that the tachyarrhythmia reduction after atropine administration could be due to a cardiac conduction improvement. According to this, a clinical report showed that atropine was also used to treat a “spice”-intoxicated patient with cardiac dysfunction [16].

Regarding PD, results showed an increase after atropine administration, differently from what was expected based on the evidence. Indeed, atropine should block the peripheral vasodilation, preventing the acetylcholine action [83], and this was confirmed by in vivo studies on dogs [84,85]. Nevertheless, the pulse distension increase is consistent with Abraham and colleagues’ study [86], which showed the hypotensive effect of atropine in hypertensive rats. Actually, these findings demonstrated that atropine can modulate noradrenergic system response and this action underlies its hypotensive effect [86]. This has probably been a physiological reflex of mice due to an already existent high sympathetic tone [86]. An excessive sympathetic activity to the cardiovascular system may paradoxically activate cardiac sensory nerves in the vagus nerve, causing reflex inhibition of sympathetic activity to blood vessels, leading to vasodilation [87].

Vagal innervation could be involved in the JWH-018-induced depressant breathing effect [88,89]. The central effect of atropine could solve this effect. Moreover, the atropine could also interfere with the oxygen blood saturation, which further decreases the JWH-018-induced effect, during the last hour after atropine administration [90].

### 3.3. Nifedipine

Nifedipine is a Ca^2+^ channel blocker that is commonly used to treat hypertensive emergencies and as antianginal medication [91]. Despite this, administration of 1 mg/kg of nifedipine was unexpectedly inefficient in reverting the effect of JWH-018 in pulse distension values (Figure 5B). Nifedipine was also ineffective on bradycardia induced by JWH-018 even if it prevented the sudden increase of HR (Figure 6A) and the tachyarrhythmic events during the last hours (Figure 6E,F). Moreover, nifedipine was able to restore the JWH-018-induced breath rate and SpO2 reduction, immediately after administration (Figure 5C,D).

Nifedipine was used to try to manage a common effect described in many case reports of cannabinoid intoxication: hypertension. Despite nifedipine increasing PD by itself, it did not revert the effect induced by JWH-018 in mice during all experiments, except during the first hour, and the effect was confirmed by BP-2000 blood pressure analysis. This evidence could suggest that hypertension induced by JWH-018 was probably caused by a peripheral action that involves calcium channel [92] during the first hour, while during the following hours of the experiment the central action on dorsal periaqueductal gray (dPAG) could prevail [93]. However, the possible variability factors related to the technique applied (such as heat, restraint of the mouse, and inflation of the cuff on the tail) should be considered [94].

In line with the action mechanism of nifedipine on the calcium channel [91], the obtained results may suggest a possible involvement of these latter for the onset of arrhythmias [95,96], as previously hypothesized [21].

Breathing parameters of JWH-018 were slightly reverted immediately after nifedipine injection. This is in accordance with previous in vitro and in vivo studies, which showed the nifedipine through Ca^2+^ channel block within airway smooth cells, diminished airway resistance, leading to increase of breathing frequency [97,98,99,100]. Moreover, in line with our results on the SpO2 parameter, a study carried out by Watanabe and colleagues showed that nifedipine was also able to increase oxygen saturation level in hypoxic rats [101].

### 3.4. Propranolol

As a last attempt to solve both the JWH-018-induced cardiac effects and the possible onset of hypertension, it was decided to administer propranolol, a β-blocker commonly used as both an antihypertensive and antiarrhythmic drug [102]. The results showed that injection of 2 mg/kg of propranolol caused a further decrease in HR during first hours, but it reverted the HR at the end of the experiment, and it dropped tachyarrhythmic events immediately after injection (Figure 7A and Figure 8C–F, respectively). On the contrary, JWH-018-induced breath rate reduction was partially increased immediately after propranolol administration and it was further reduced at the end of the experiment (Figure 7C). Moreover, propranolol increased the SpO_2_ values (Figure 7D). The pulse distension was not affected by propranolol (Figure 7B).

In the heart, as expected, propranolol initially further decreased the low cardiac frequency of JWH-018-treated mice, causing a further increase of vagal activity due to β receptor block [102,103,104]. Concerning heart rate, treatment with propranolol induced an effect that could depend on the basal tone of JWH-018-injected mice, similar to the atropine effect on pulse distension. The increment of already high vagal tone in mice probably has led to a paradoxical effect. Indeed, following the initial and expected further decrease of heart rate, our results showed an increased HR during the last hours of the experiment, and this evidence was already shown in a clinical study [105]. Moreover, the reduction of tachyarrhythmia onset was in line with clinical studies with the well-known evidence that propranolol reduces arrhythmias, especially supraventricular tachyarrhythmias [106,107,108,109]. In particular, propranolol exerts its action on cardiac β1 receptors [109], leading to the reduction of ventricular contraction through catecholamines block on A-V node and hence could reduce and improve the chronotropic cardiac effect on mice [106]. The diminution of cardiac output, as well as the central and peripheral reduction of the sympathetic tone caused by propranolol [110], should suggest an increase in pulse distension. Despite the cardiac action, reduction of pulse distension induced by JWH-018 was not modulated by propranolol administration and this could suggest the hypothesis to exclude the involvement of sympathetic nerves from JWH-018-induced hypertension. Moreover, the breath rate trend could be related to heart failure. The effect could be closely linked to a probable JWH-018-induced ventricular dysfunction that could lead to respiratory damage [111,112]. Indeed, in heart failure patients, β blockers were able to improve pulmonary hemodynamics, diminishing the liquid content of the lung tissue and its relative effects on bronchial, alveolar, and interstitial tissue [113]. This could suggest why propranolol initially reverted JWH-018-induced bradypnea. Moreover, propranolol was able to abolish oxygen saturation reduction in accordance with Khambatta and colleagues’ preclinical study [114]. The decrease of sympathetic tone could then prevail, further decreasing breath rate [115].

Our study is limited to the use of male mice and this choice was made based on increasing evidence of the greater risk of consumption among male adults than female and the higher susceptibility of males to SCs effects has been shown. In fact, previous reports have shown emergency room assistance has been required more frequently for males (78%) than females (22%) patients following SCs intoxication in respect to cannabis intoxication [116]. Moreover, males have accounted for the 73.9% of patients who have contacted poison centers after SCs use and reported tachycardia among main adverse effects [14]. In line with these findings, Fogel and colleagues have more recently demonstrated that women can be less sensitive to the effects induced by high dosages of THC [117]. Despite this, both clinical [118] and preclinical [119,120] evidence suggest that females are more susceptible to cannabinoid-induced effects, even though cardiovascular responses are not mentioned among these studies.

Overall, our study sheds light on the CV effects of four pharmacological agents that are commonly used in Emergency Departments. In particular, the use of amiodarone and propranolol, and possibly other beta-blockers that must be tested, are particularly interesting because both of them can be used in the presence of coronary artery diseases and acute myocardial infarction. A systematic review [121] including 115 studies and other reports [19] has proven the increased risk of cardiovascular diseases, acute myocardial infarction, and ischemic stroke in healthy and young people consuming cannabis and synthetic cannabiminetics, so it will be pivotal to translate these findings to humans. In fact, the presence of tachycardia is the most reliable marker to study the effects of cannabinoids in humans [122] but it is also a predictor of an increased risk of CV morbidity and mortality because it leads to a reduction in the cardiac stroke volume and impairs the myocardial oxygen supply–demand. HR reduction is therefore a protective effect in the presence of acute myocardial infarction alone, and also in cannabinoid-induced tachycardia and their synergic effects could be particularly detrimental.

## 4. Materials and Methods

### 4.1. Animals

Male outbred ICR (CD-1^®^) mice (N = 112), 25–30 g (Centralized Preclinical Research Laboratory, University of Ferrara, Italy) were group-housed (five mice per cage; floor area per animal was 80 cm^2^; minimum enclosure height was 12 cm) in a colony room under a constant temperature (23–24 °C) and humidity (45–55%). Food (Diet 4RF25 GLP; Mucedola, Settimo Milanese, Milan, Italy) and tap water were available ad libitum during the entire time the animals spent in their home cages. The daylight cycle was artificially maintained (dark between 7 p.m. and 7 a.m.). The experiments were performed during the light phase. The experimental protocol followed in the present study was in accordance with the new European Communities Council Directive of September 2010 (2010/63/EU), a revision of the Directive 86/609/EEC, and was approved by the Italian Ministry of Health (license no. 223/2021-PR and extension CBCC2.46.EXT.21) and the Ethics Committee of the University of Ferrara. According to the ARRIVE guidelines, all possible efforts were made to minimize the number of animals used, minimize the animals’ pain and discomfort, and reduce the number of experimental subjects. In this study, the determination of the number of animals to be used (sample size) and the calculation of the appropriate power in the statistical data analysis (power analysis) was determined using the simulation software G*Power 3.1.9.2 (Heinrich-Heine-Universität Düsseldorf, Düsseldorf, Germany) [123]. Following the manual G*Power 3.1.9.2 we then carried out the Prior power analysis [124] that allows calculation of the number of animals to be used (N or sample size). Thus, in function of the actual power level of the analysis (0.99 < β < 1.00) of the level of significance (α = 0.05) to be achieved and according to the magnitude of the effect reported in the present tests (effect size f) a sample size of eight animals per group was calculated. In the cardiorespiratory studies (vehicle, JWH-018 6 mg/kg, amiodarone 5 mg/kg, atropine 5 mg/kg, nifedipine 1 mg/kg, propranolol 2 mg/kg, JWH-018 + amiodarone, JWH-018 + atropine, JWH-018 + nifedipine, or JWH-018 + propranolol) eight mice were used per group (total mice used: 80). For systolic and diastolic pressure study (saline, JWH-018 6 mg/kg, nifedipine, and JWH + nifedipine) eight mice were used per group (total mice used: 32). For all experiments, only male mice were used, following international trends that identify men as the main cannabinoid consummators [125].

### 4.2. Drug Preparation and Dose Selection

JWH-018 was purchased from LGC Standards (LGC Standards S.r.L., Sesto San Giovanni, Milan, Italy) while amiodarone, atropine, nifedipine, and propranolol were from Tocris (Tocris, Bristol, UK). Drugs were initially dissolved in absolute ethanol (final concentration was 5%) and Tween 80 (2%) and brought to the final volume with saline (0.9% NaCl). The solution made with ethanol, Tween 80, and saline was also used as the vehicle. Amiodarone (5 mg/kg), atropine (5 mg/kg), nifedipine (1 mg/kg), and propranolol (2 mg/kg) were administered 60 min after JWH-018 injection. Drugs were administered by intraperitoneal injection (i.p.) in a volume of 4 µL/g.

The dose selection was evaluated based on a previous study by our group [21] that showed severe CV effects induced by 6 mg/kg JWH-018. In line with HED formula and with dosage scale reported by users [126,127], this dose represents a toxic dose of JWH-018 in humans. The dosage of amiodarone [128], atropine [129,130], nifedipine [131], and propranolol [132] were chosen from previous preclinical studies on rodents.

### 4.3. Evaluation of Cardiorespiratory Changes

As previously reported, to monitor the cardiorespiratory parameters in awake and freely moving animals without using invasive instruments and handling, a collar with a sensor was used to detect heart rate (HR), breath rate (BR), oxygen blood saturation (SpO2), and pulse distension (PD) with a frequency of 15 Hz [133,134,135,136]. During the experiment, the mouse was allowed to freely move around its cage (30 × 30 × 20 cm) while having no access to food or water while being monitored by the sensor collar through the software MouseOx Plus 1.6 (STARR Life Sciences^®^ Corp., Oakmont, PA, USA). In the first hour of acclimation, a fake collar similar in design to the collar used in the test but without a sensor was used to minimize the potential stress of the mouse during the experiment. The collar with the sensor was then applied, while the baseline parameters were monitored for 60 min. Subsequently, drugs or the vehicle were administered. Amiodarone (5 mg/kg), atropine (5 mg/kg), nifedipine (1 mg/kg), and propranolol (2 mg/kg) were administered 60 min after JWH-018 injection. The data were recorded for 5 h.

Due to its primary effects on blood pressure, the co-administration with nifedipine (1 mg/kg) is also evaluated on BP-2000 system. As previously reported [133], systolic and diastolic blood pressure were measured by tail-cuff plethysmography using a BP-2000 blood pressure analysis system (Visitech Systems, Apex, NC, USA). For each session, mice were placed in a metal box restraint with its tail passing through the optical sensor and compression cuff before finally being taped to the platform. A traditional tail-cuff occluder was placed proximally on the animal’s tail, which was then immobilized with tape in a V-shaped block between a light source above and a photoresistor below. Upon inflation, the occluder stopped blood flow through the tail, while upon deflation, the sensor detected the blood flow return. The restraint platform was maintained at 37 °C. Before experiments, mice were acclimated to restraint and tail-cuff inflation for 5–7 days. On the test day, 10 measurements were made to collect basal blood pressure. Upon the tenth analysis, the software was paused, and mice were injected with either drug treatments or the vehicle; animals were then repositioned in the restraints, and 60 measurements were acquired.

### 4.4. Data and Statistical Analysis

Data related to HR, PD, BR, and SpO2 changes are expressed as a percentage of basal value. While tachyarrhythmia analysis expressed in histograms represents the number of tachyarrhythmia events (divided in each hour, for 6 h). A tachyarrhythmic event was recorded when heart pulse was almost >200 pulses higher compared to mean basal HR, after vehicle or drug administration [21]. The statistical analysis of the dose–response curve of different substances and the analysis of tachyarrhythmia frequencies was performed by two-way ANOVA followed by Bonferroni’s test for multiple comparisons.

The significance level was set at *p* < 0.05. Cardiovascular data are expressed as a percentage of the baseline value with mean ± standard error of the mean (SEM) of the eight independent experiments. Tachyarrhythmia data are expressed as a number of events with mean ± SEM of the eight independent experiments. All statistical analyses were performed using GraphPad Prism 8.0.1 software (GraphPad Prism, San Diego, CA, USA). Changes in systolic and diastolic blood pressure were expressed as absolute values (mmHg) of average effect. The effects of different average effect of each substance were analyzed by a one-way ANOVA followed by Bonferroni’s test for multiple comparisons where appropriate. Data were reported as mean standard error of the mean (SEM) of at least eight independent experiments.

## 5. Conclusions

In conclusion, our study helps to better clarify the mechanisms under cardio-respiratory dysfunctions induced by JWH-018 (6 mg/kg). In particular, our results corroborated a previous study hypothesis [21] in which heart rate decrease was linked to vagal tone increase via CB1 or CB2 receptors. Indeed, only muscarinic receptor block, through atropine administration, entirely enhanced JWH-018-induced bradycardia one hour from administration. Beyond this, our results showed that all examined CV drugs improved tachyarrhythmias induced by JWH-018, suggesting different mechanisms behind onset of tachyarrhythmias, including vagal and sympathetic tone disbalance, and ion channel involvement. Regarding pulse distension, only atropine slightly increased it, but as above explained, this could be a reflex mechanism. Finally, even breathing responses suggested different mechanisms involved. Again, all drugs tested improved respiratory rate, in particular amiodarone administration. These data, beyond suggesting and defining different mechanisms behind JWH-018-induced CV damage, are important to identify possible antidotes in case of JWH-018 or other SCs cardiac intoxication, highlighting the strong public health impact induced by JWH-018 diffuse use.

## Figures and Tables

**Figure 1 ijms-24-07515-f001:**
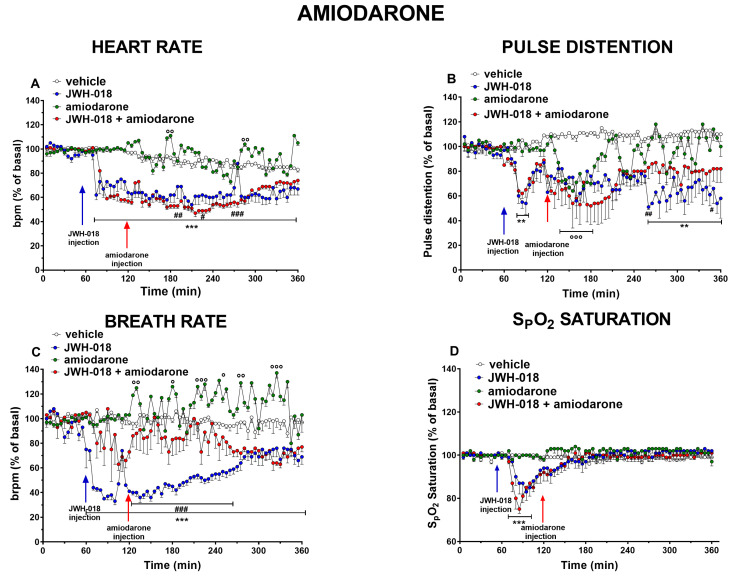
Effect of systemic administration of JWH-018 (6 mg/kg), amiodarone (5 mg/kg), and JWH-018 followed by amiodarone on heart rate (**A**), pulse distention (**B**), breath rate (**C**), and arterial oxygen saturation (**D**). Data are expressed as percentage of basal values in the form mean ± SEM of eight different evaluations for each group. Statistical analysis was performed by two-way ANOVA followed by Bonferroni’s test for multiple comparisons. ** *p* < 0.01, *** *p* < 0.001, JWH-018 versus vehicle. # *p* < 0.05, ## *p*< 0.01, ### *p*< 0.001, JWH-018 versus JWH-018 + amiodarone. ° *p* < 0.05, °° *p* < 0.01, °°° *p* < 0.001, amiodarone versus vehicle.

**Figure 2 ijms-24-07515-f002:**
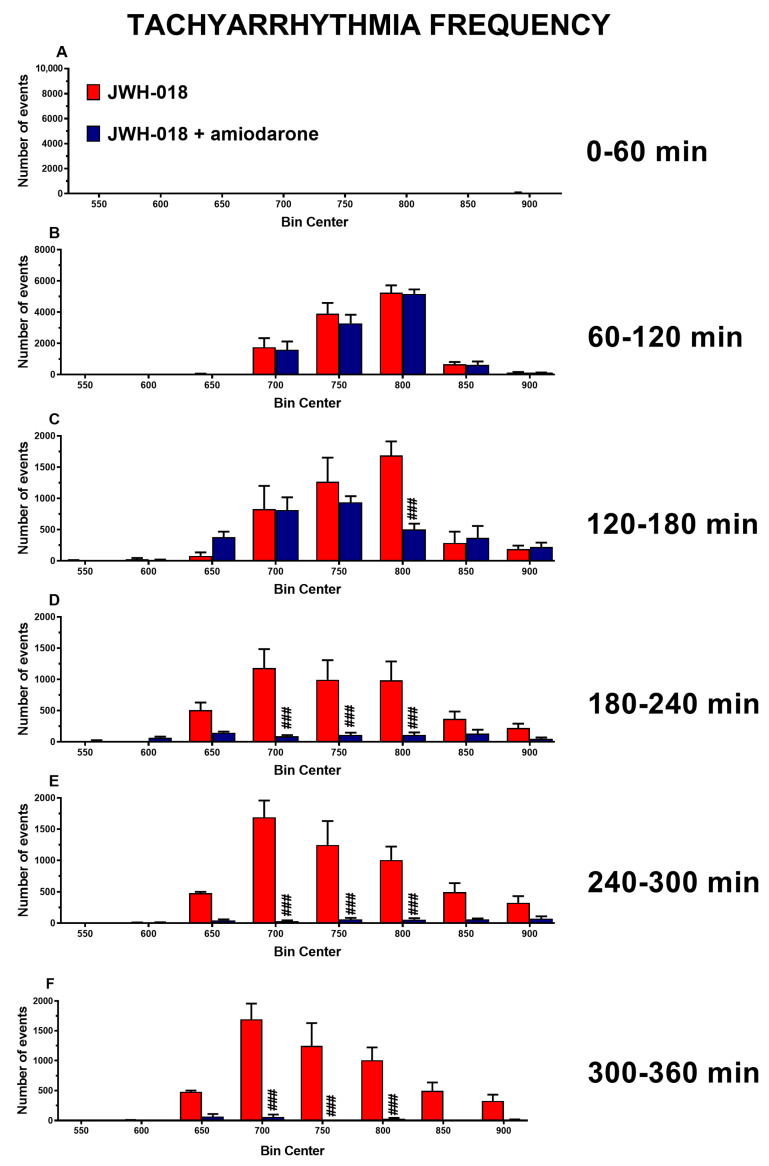
Frequency of tachyarrhythmia episodes between 0–60 (**A**), 60–120 (**B**), 120–180 (**C**), 180–240 (**D**), 240–300 (**E**), and 300–360 (**F**) minutes after administration of JWH-018 (6 mg/kg) or JWH-018 followed by amiodarone (5 mg/kg), expressed as number of events per mean heart rate value. Mean ± SEM of eight different evaluations for each group. Statistical analysis was performed by two-way ANOVA followed by Bonferroni’s test for multiple comparisons. ### *p* < 0.001 versus JWH-018 + amiodarone.

**Figure 3 ijms-24-07515-f003:**
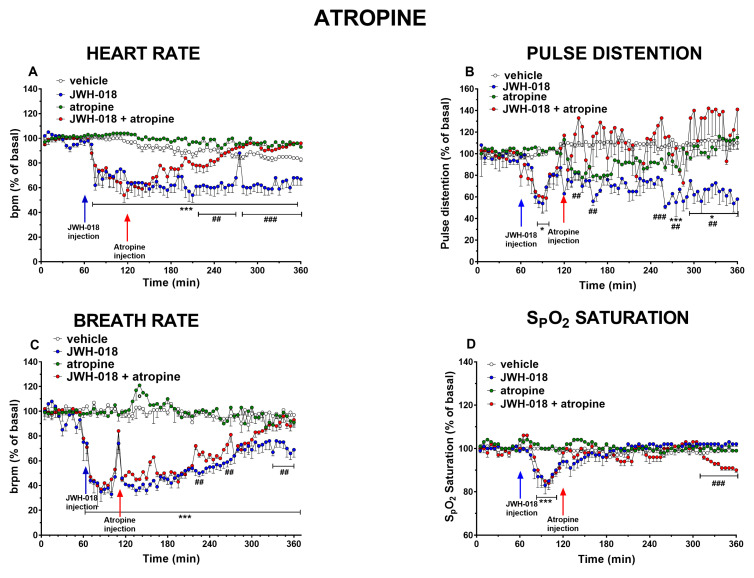
Effect of systemic administration of JWH-018 (6 mg/kg), atropine (5 mg/kg), or JWH-018 followed by atropine on heart rate (**A**), pulse distention (**B**), breath rate (**C**), and arterial oxygen saturation (**D**). Data are expressed as percentage of basal values in the form mean ± SEM of eight different evaluations for each group. Statistical analysis was performed by two-way ANOVA followed by Bonferroni’s test for multiple comparisons. * *p* < 0.05, *** *p* < 0.001, JWH-018 versus vehicle. ## *p* < 0.01, ### *p*< 0.001 versus JWH-018 + atropine. ° *p* < 0.05, atropine versus vehicle.

**Figure 4 ijms-24-07515-f004:**
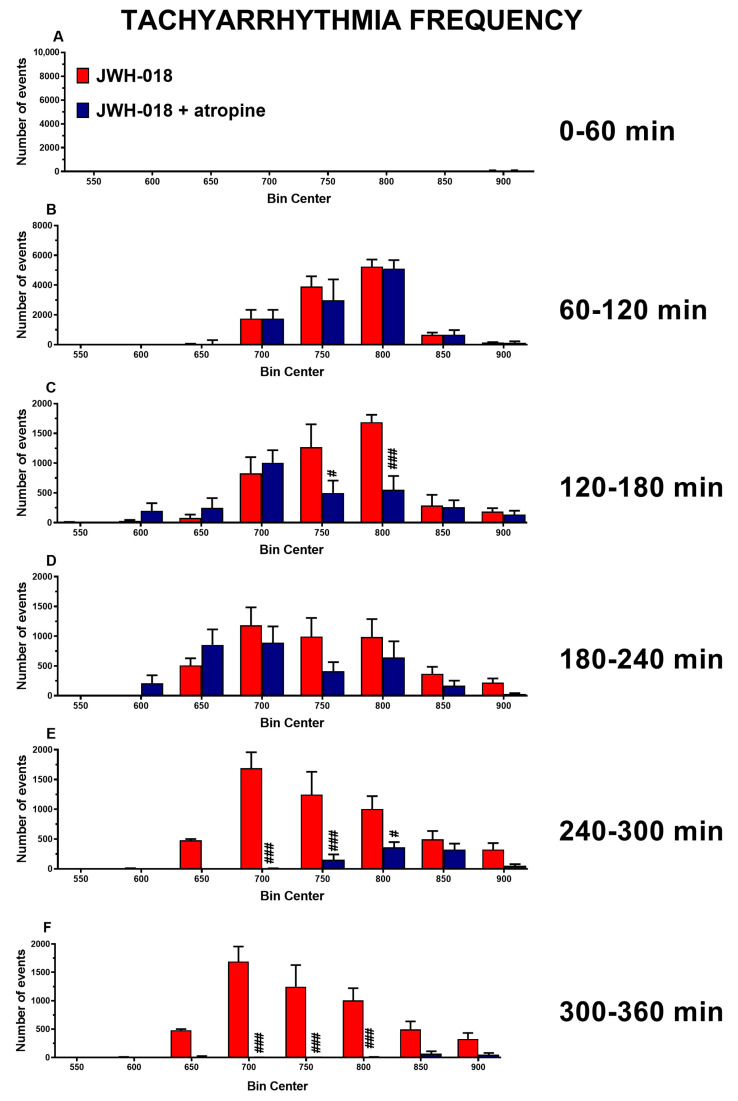
Frequency of tachyarrhythmia episodes between 0–60 (**A**), 60–120 (**B**), 120–180 (**C**), 180–240 (**D**), 240–300 (**E**), and 300–360 (**F**) minutes after administration of JWH-018 (6 mg/kg) or JWH-018 followed by atropine (5 mg/kg), expressed as number of events per mean heart rate value. Mean ± SEM of eight different evaluations for each group. Statistical analysis was performed by two-way ANOVA followed by Bonferroni’s test for multiple comparisons. # *p* < 0.05, ### *p*< 0.001 versus JWH-018 + atropine.

**Figure 5 ijms-24-07515-f005:**
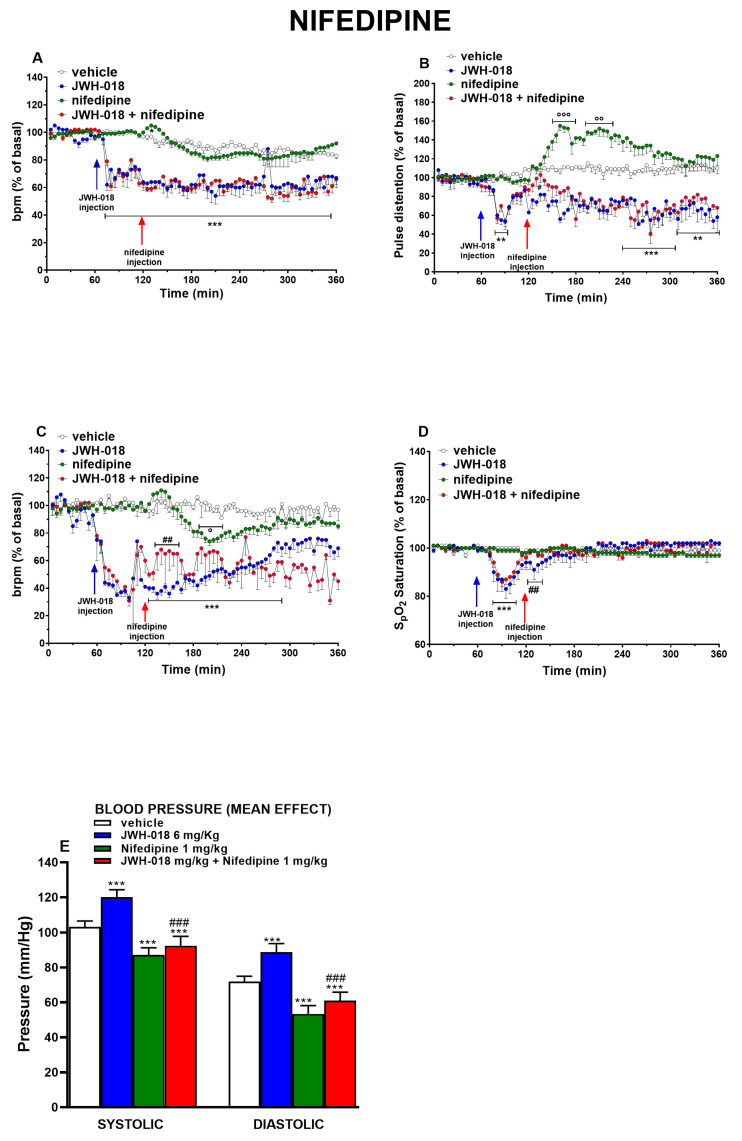
Effect of systemic administration of JWH-018 (6 mg/kg), nifedipine (1 mg/kg), or JWH-018 followed by nifedipine on heart rate (**A**), pulse distention (**B**), breath rate (**C**), and arterial saturation (**D**). Data are expressed as percentage of basal values in the form Mean ± SEM of eight different evaluations for each group. Statistical analysis was performed by two-way ANOVA followed by Bonferroni’s test for multiple comparisons. ** *p* < 0.01, *** *p* < 0.001, JWH-018 versus vehicle. ## *p* < 0.01, JWH-018 versus JWH-018 + nifedipine. ° *p* < 0.05, °° *p* < 0.01, °°° *p* < 0.001, JWH-018 nifedipine versus vehicle. Changes in systolic and diastolic blood pressure (**E**). Data are expressed as absolute values (mmHg) of average effect. Statistical analysis was performed by one-way ANOVA followed by Bonferroni’s test for multiple comparisons. *** *p* < 0.001 versus vehicle. ## *p* < 0.01 JWH-018, ### *p* < 0.001 versus JWH-018 + nifedipine.

**Figure 6 ijms-24-07515-f006:**
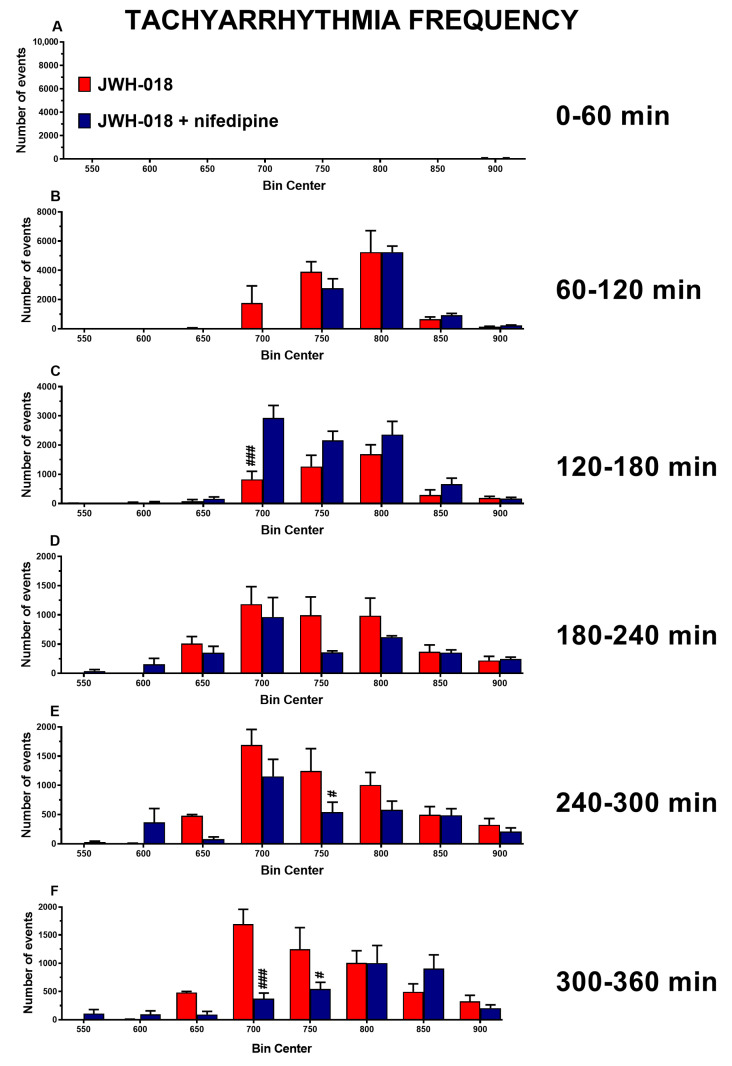
Frequency of tachyarrhythmia episodes between 0–60 (**A**), 60–120 (**B**), 120–180 (**C**), 180–240 (**D**), 240–300 (**E**), and 300–360 (**F**) minutes after administration of JWH-018 (6 mg/kg) or JWH-018 followed by nifedipine (5 mg/kg), expressed as number of events per mean heart rate value. Mean ± SEM of eight different evaluations for each group. Statistical analysis was performed by two-way ANOVA followed by Bonferroni’s test for multiple comparisons. # *p* < 0.05, ### *p* < 0.001 versus JWH-018 + nifedipine.

**Figure 7 ijms-24-07515-f007:**
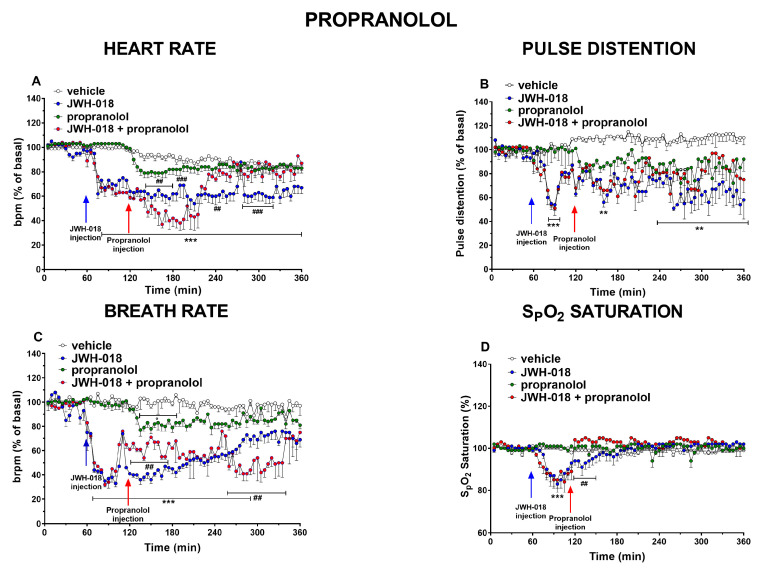
Effect of systemic administration of JWH-018 (6 mg/kg), propranolol (2 mg/kg), or JWH-018 followed by propranolol on heart rate (**A**), pulse distention (**B**), breath rate (**C**), and oxygen blood saturation (**D**). Data are expressed as percentage of basal values in the form mean ± SEM of four different evaluations for each group. Statistical analysis was performed by two-way ANOVA followed by Bonferroni’s test for multiple comparisons. ** *p* < 0.01, *** *p* < 0.001, JWH-018 versus vehicle. ## *p* < 0.01, ### *p*< 0.001, JWH-018 versus JWH-018 + propranolol. ° *p* < 0.05, propranolol versus vehicle.

**Figure 8 ijms-24-07515-f008:**
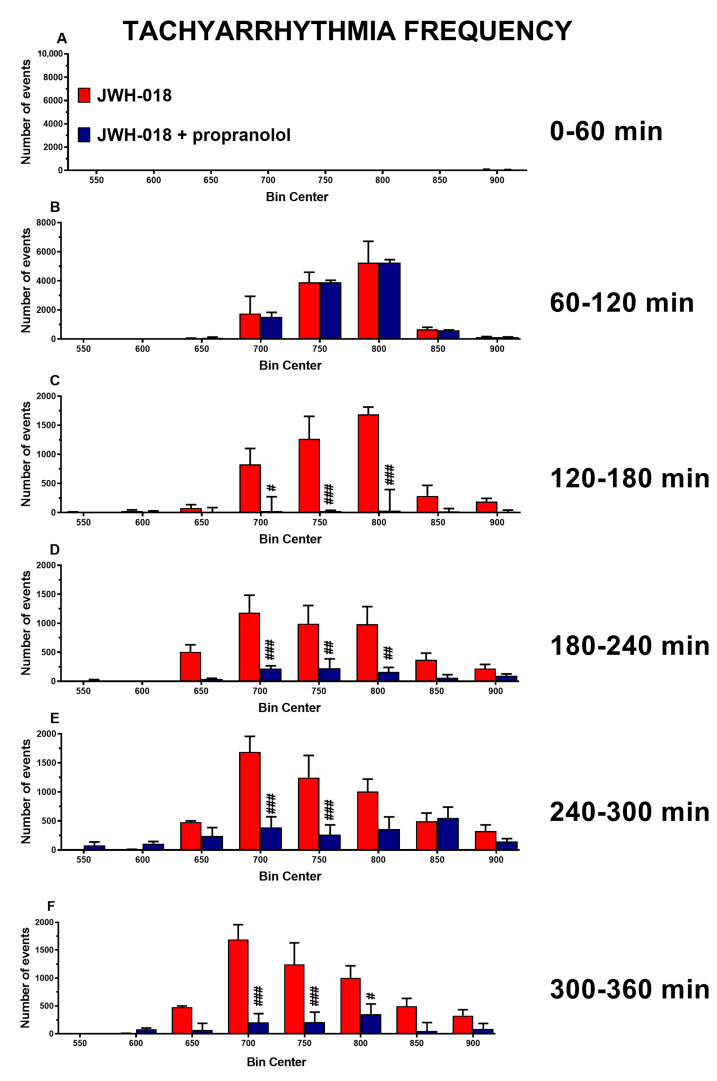
Frequency of tachyarrhythmia episodes between 0–60 (**A**), 60–120 (**B**), 120–180 (**C**), 180–240 (**D**), 240–300 (**E**), and 300–360 (**F**) minutes after administration of JWH-018 (6 mg/kg) or JWH-018 followed by propranolol (2 mg/kg), expressed as number of events per mean heart rate value. Mean ± SEM of eight different evaluations for each group. Statistical analysis was performed by two-way ANOVA followed by Bonferroni’s test for multiple comparisons. # *p* < 0.05, ## *p* < 0.01, ### *p* < 0.001 versus JWH-018 + propranolol.

## Data Availability

The data presented in this study are available on request from the first author (Beatrice Marchetti) and corresponding author (Matteo Marti) for researchers of academic institutes who meet the criteria for access to the confidential data.

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
