# Peer review of "Acute Cardiovascular and Cardiorespiratory Effects of JWH-018 in Awake and Freely Moving Mice: Mechanism of Action and Possible Antidotal Interventions?"

_ijms, 2023, doi:10.3390/ijms24087515_

Round 1

Reviewer 1 Report

This is a very interesting study highlighting the dangerous CV side effects of synthetic cannabinoids. In light of the public health crisis with substance abuse, stemming partially from the pandemic, this is an important public health concern. The overall study was designed adequately, please find below some comments that in my opinion would improve the manuscript:

1. While the introduction contains a plethora of relevant information, perhaps it would be useful to denote which data comes from mouse and which data from human studies. Or for example, it could be useful to have statement of sort that compares the rodent and human brain in terms of the endocannabinoid system and delineates any differences that are known to exist. I understand that it may be difficult simply based on the fact that the data is lacking describing such differences, but even a brief comparison between the two species would be appreciated. In my opinion, this will position your studies that are conducted in mice more relevantly to the audience who is more interested in  the translational aspect of this work.

2. Tail cuff measurements of blood pressure are not gold standard, rather telemetry is. I understand that this is unreasonable to be asking in this review. However, could you please comment on the day to day variability of BP measurements with tail cuff? 

3. The power analysis in unclear: what was the effect size, what was the level of significance pre-specified and what was the actual power at these parameters?

4. Please include the rationale for including only male mice. I understand that you reference that in humans, males are consuming cannabinoids more than females. However, females still do consume cannabinoids. Perhaps include some discussion on whether, and how, the existing literature points to the potential different responses in females compared to males.

Author Response

Response to Reviewer 1

We thank the Reviewer 1 for his/her evaluation of our manuscript and for helpful concerns to improve the article. In this revised version of the work we have addressed the major concerns of the referee (highlighted in green).

This is a very interesting study highlighting the dangerous CV side effects of synthetic cannabinoids. In light of the public health crisis with substance abuse, stemming partially from the pandemic, this is an important public health concern. The overall study was designed adequately, please find below some comments that in my opinion would improve the manuscript:

Rev1Q1: While the introduction contains a plethora of relevant information, perhaps it would be useful to denote which data comes from mouse and which data from human studies. Or for example, it could be useful to have statement of sort that compares the rodent and human brain in terms of the endocannabinoid system and delineates any differences that are known to exist. I understand that it may be difficult simply based on the fact that the data is lacking describing such differences, but even a brief comparison between the two species would be appreciated. In my opinion, this will position your studies that are conducted in mice more relevantly to the audience who is more interested in the translational aspect of this work.

AA: We thank the Reviewer 1 for this comment and provide a new version of the introduction, in which we have specified which data comes from mouse and which data from human studies to briefly highlight interspecies similarities and disparities.

Rev1Q2: Tail cuff measurements of blood pressure are not gold standard, rather telemetry is. I understand that this is unreasonable to be asking in this review. However, could you please comment on the day to day variability of BP measurements with tail cuff?

AA: We thank the Reviewer 1 for highlighting this interesting point. As scientific literature about the topic states, telemetry is considered as a gold standard technique for accurate blood pressure evaluation. However, the costs and complexity of the surgical implant necessary for the implementation of this technique should be considered. In particular, the application of blood pressure measurements by tail cuff may be considered appropriate in this experimental session. In fact, Feng and colleagues have previously highlighted that this type of measurement is suitable for evaluating models of moderate hypertension (110-180 mmHg). Unlike telemetry measurements under these specific conditions, the tail cuff technique did not show increased mortality. Finally, the latter appears to mimic the models applied in the clinical setting confirming the value of this method with reference to the potential translational settings of the present study. However, the heat and the restrain of animals might be the subject of variability in the BP measurements with tail cuff and to reduce the variability animals were trained for 5 to 7 days as mentioned in materials and methods. BP control measurements were analyzed and the animals showing very different BP values were excluded to reduce variability in this test. Despite these two main sources of variability in BP measurements by the tail cuff, this technique was showing high reproducibility and strong correlation with BP-intra-arterial in resting mice (Krege et al.,1995).

We included the comment on greater variability linked to tail cuff measurement in the manuscript as suggested by the Reviewer 1.

Feng M, Whitesall S, Zhang Y, Beibel M, D'Alecy L, DiPetrillo K. Validation of volume-pressure recording tail-cuff blood pressure measurements. Am J Hypertens. 2008;21(12):1288-1291. doi:10.1038/ajh.2008.301

Luther JM, Fogo AB. Under pressure-how to assess blood pressure in rodents: tail-cuff?. Kidney Int. 2019;96(1):34-36. doi:10.1016/j.kint.2018.12.034

Fink GD. Does Tail-Cuff Plethysmography Provide a Reliable Estimate of Central Blood Pressure in Mice? J Am Heart Assoc. 2017 Jun 27;6(6):e006554. doi: 10.1161/JAHA.117.006554. PMID: 28655736; PMCID: PMC5669206.

Krege, J. H., Hodgin, J. B., Hagaman, J. R., & Smithies, O. (1995). A noninvasive computerized tail-cuff system for measuring blood pressure in mice. Hypertension (Dallas, Tex. : 1979), 25(5), 1111–1115. https://doi.org/10.1161/01.hyp.25.5.1111

Rev1Q3: The power analysis is unclear: what was the effect size, what was the level of significance pre-specified and what was the actual power at these parameters?

AA: We thank the Reviewer 1 for this comment and we have modified the specific section of the manuscript in the effort to clarify this point.

Rev1Q4: Please include the rationale for including only male mice. I understand that you reference that in humans, males are consuming cannabinoids more than females. However, females still do consume cannabinoids. Perhaps include some discussion on whether, and how, the existing literature points to the potential different responses in females compared to males.

AA: We thank the Reviewer 1 for this comment. We believe that incorporating sex differences to study the possible antidotal interventions for cardiotoxicity induced by synthetic cannabinoids is very important. The main reason to use only males in our study is because most case reports with JWH-018 are related to males and we have showed in the above-mentioned review (Fattore et al., 2020) that males might be more susceptible to the adverse effects of SCRAs. Moreover, many studies have evidence on the role of the gonadal hormones on the regulation of the endocannabinoid system both in preclinical (Rodriguez de Fonseca et al, 1994; Castelli et al., 2014) and clinical (El-Talatini et al, 2010) studies. Indeed, it has been demonstrated that the pharmacokinetics and pharmacodynamics of the cannabinoids is influenced in females by Oestradiol (Wakley et al, 2014; Craft et al., 2017). We are planning on performing a future study on female mice to add more evidence on sex differences related to cardiotoxicity following exposure to the synthetic cannabinoid JWH-018.

We added information about the topic in the specific section of the manuscript in which we have indicated the use of only male mice.

References:

Fattore, L., Marti, M., Mostallino, R., & Castelli, M. P. (2020). Sex and Gender Differences in the Effects of Novel Psychoactive Substances. Brain sciences, 10(9), 606. https://doi.org/10.3390/brainsci10090606

Rodriguez de Fonseca F, Cebeira M, Ramos JA, Martín M, Fernández-Ruiz JJ (1994). Cannabinoid receptors in rat brain areas: sexual differences, fluctuations during estrous cycle and changes after gonadectomy and sex steroid replacement. Life Sci 54: 159–170.

Castelli MP, Fadda P, Casu A, Spano MS, Casti A, Fratta W et al (2014). Male and female rats differ in brain cannabinoid CB1 receptor density and function and in behavioural traits predisposing to drug addiction: effect of ovarian hormones. Curr Pharm Des 20: 2100–2113.

El-Talatini MR, Taylor AH, Konje JC (2010). The relationship between plasma levels of the endocannabinoid, anandamide, sex steroids, and gonadotrophins during the menstrual cycle. Fertil Steril 93: 1989–1996.

Wakley AA, McBride AA, Vaughn LK, Craft RM (2014). Cyclic ovarian hormone modulation of supraspinal Δ9-tetrahydrocannabinol-induced antinociception and cannabinoid receptor binding in the female rat. Pharmacol Biochem Behav 124C: 269–277.

Craft RM, Haas AE, Wiley JL, Yu Z, Clowers BH (2017). Gonadal hormone modulation of Δ9-tetrahydrocannabinol-induced antinociception and metabolism in female versus male rats. Pharmacol Biochem Behav 152: 36–43. Maldonado R, Cabañero D, Martín-García E. The endocannabinoid system in modulating fear, anxiety, and stress. Dialogues Clin Neurosci. 2020;22(3):229-239. doi:10.31887/DCNS.2020.22.3/rmaldonado

.

Reviewer 2 Report

Privous study showed that higher doses of the synthetic cannabinoid JWH-018 (3-6 mg/kg) induced deep and long-lasting bradycardia, alternated with bradyarrhythmia, spaced out by sudden episodes of tachyarrhythmias (6 mg/kg), and characterized by ECG electrical parameters changes, sustained bradypnea, and systolic and transient diastolic hypertension in awake CD-1 male mice. These effects were prevented by both treatment with selective CB1 (AM 251, 6 mg/kg) and CB2 (AM 630, 6 mg/kg) receptor antagonists. Cardio-respiratory and vascular symptoms could be induced by peripheral and central CB1 and CB2 receptors stimulation, which could lead to both sympathetic and parasympathetic systems activation. The present study investigated how cardio-respiratory and -vascular responses of JWH-018 (6 mg/kg) can be modulated by antidotes al-ready in clinical use, including amiodarone (5 mg/kg), atropine (5 mg/kg), nifedi-pine (1 mg/kg), and propranolol (2 mg/kg). The detection of heart rate, breath rate, arterial oxygen blood saturation (SpO2), and pulse distention were provided by a non-invasive apparatus (Mouse Ox Plus) in awake and freely moving CD-1 male mice. Tachyarrhythmias events are also evaluated. Results showed that only atropine completely revert the reduction of heart rate and pulse distension; all tested antidotes reduce tachycardia and tachyarrhythmias events and improve breathing functions. Some concerns such as typos and errors are listed in the following:

L44: derivates? Derivatives

*L98-99: 5.81+ 0.02, 5.88+0.03, 99 and 5.37+0.02? -> concentration unit?

*L30, L127, 185: arterial saturation?

**L173: it enhanced? the bradycardic effect induced by JWH-018 during the last three hours of the experiment -> reverted

*L220: ## p<0.01 J><0.01-> but in (D), ### was labeled

L255: β1 β2 blocker

**L350-351: enhanced? the JWH-018-induced vasoconstriction -> reverted

L357, L371: K+, Na+, Ca2+

*L380: to contrast tachycardia; L389: contrast the action; L399: contrast JWH-018-induced CV effects; L418: contrasting the acetylcholine action; L434: contrasting the action and so on -> the word ‘contrast’ is not often used

**L421-425: This effect has been probably a physiological reflex of mice due to an already existent high sympathetic tone [85]. An excessive sympathetic activity to the cardiovascular system may paradoxically activate cardiac sensory nerves in the vagus nerve, causing reflex inhibition of sympathetic activity to blood vessels, leading to vasodilation [86]. -> atropine can block this reflex activation of vagus nerve, then it should lead to vasoconstriction. Actually, in Ref 85, it demonstrated that atropine acts as a competitive antagonist of norepinephrine and this action underlies its hypotensive effect.

*L428: the atropine metabolism could also interfere with the oxygen blood saturation -> Is it demonstrated in Ref 89?

L432, L453: Ca2+ channel

*L485: Despite the cardiac action, reduction of pulse distension induced by JWH-018 did not enhance? after propranolol (possible a but not b adrenergic effect)

L512: 25-30 gr? L547: 4 µl/gr -> gram

L514: 80 cm2

*L643-958: Abbreviations: It is better to list in alphabetic order

*L659-: References (check all): inconsistent writing format for

(1) title: capital letter on the first word only (ref 6) vs. all words (ref 25)

(2) page number: ref 7: 1251-1261 vs. ref 11: 234-43

Author Response

Response to Reviewer 2

We thank the Reviewer 2 for his/her evaluation of our manuscript and for helpful concerns to improve the article. In this revised version of the work we have addressed the major concerns of the referee (highlighted in blue).

Previous study showed that higher doses of the synthetic cannabinoid JWH-018 (3-6 mg/kg) induced deep and long-lasting bradycardia, alternated with bradyarrhythmia, spaced out by sudden episodes of tachyarrhythmias (6 mg/kg), and characterized by ECG electrical parameters changes, sustained bradypnea, and systolic and transient diastolic hypertension in awake CD-1 male mice. These effects were prevented by both treatment with selective CB1 (AM 251, 6 mg/kg) and CB2 (AM 630, 6 mg/kg) receptor antagonists. Cardio-respiratory and vascular symptoms could be induced by peripheral and central CB1 and CB2 receptors stimulation, which could lead to both sympathetic and parasympathetic systems activation. The present study investigated how cardio-respiratory and -vascular responses of JWH-018 (6 mg/kg) can be modulated by antidotes already in clinical use, including amiodarone (5 mg/kg), atropine (5 mg/kg), nifedipine (1 mg/kg), and propranolol (2 mg/kg). The detection of heart rate, breath rate, arterial oxygen blood saturation (SpO2), and pulse distention were provided by a non-invasive apparatus (Mouse Ox Plus) in awake and freely moving CD-1 male mice. Tachyarrhythmias events are also evaluated. Results showed that only atropine completely revert the reduction of heart rate and pulse distension; all tested antidotes reduce tachycardia and tachyarrhythmias events and improve breathing functions. Some concerns such as typos and errors are listed in the following:

Rev2Q1: L44: derivates? Derivatives

AA: We thank the Reviewer 2 for pointing out this inaccuracy and have corrected the sentence.

Rev2Q2: *L98-99: 5.81+ 0.02, 5.88+0.03, 99 and 5.37+0.02? -> concentration unit?

AA: We thank the Reviewer 2 for pointing out this inaccuracy. Since in the reference relative to these values, authors refers to pEC50 and not to EC50, we have corrected the sentence.

Rev2Q3: *L30, L127, 185: arterial saturation?

AA: We thank the Reviewer 2 for pointing out this inaccuracy and have corrected the sentences.

Rev2Q4: **L173: it enhanced? the bradycardic effect induced by JWH-018 during the last three hours of the experiment -> reverted

AA: We thank the Reviewer 2 for pointing out this inaccuracy and have corrected the sentence.

Rev2Q5: *L220: ## p<0.01 J><0.01-> but in (D), ### was labeled

AA: We thank the Reviewer 2 for pointing out this inaccuracy and have corrected the figure caption.

Rev2Q6: L255: β1 β2 blocker

AA: We thank the Reviewer 2 for his/her comment and have modified the subtitle.

Rev2Q7: **L350-351: enhanced? the JWH-018-induced vasoconstriction -> reverted

AA: We thank the Reviewer 2 for pointing out this inaccuracy and have corrected the sentence.

Rev2Q8: L357, L371: K+, Na+, Ca2+

AA: We thank the Reviewer 2 for pointing out this inaccuracy and have corrected the sentence.

Rev2Q9: *L380: to contrast tachycardia; L389: contrast the action; L399: contrast JWH-018-induced CV effects; L418: contrasting the acetylcholine action; L434: contrasting the action and so on -> the word ‘contrast’ is not often used

AA: We thank the Reviewer 2 for his/her comment and have modified the sentence avoiding the use of ‘contrast’.

Rev2Q10: **L421-425: This effect has been probably a physiological reflex of mice due to an already existent high sympathetic tone [85]. An excessive sympathetic activity to the cardiovascular system may paradoxically activate cardiac sensory nerves in the vagus nerve, causing reflex inhibition of sympathetic activity to blood vessels, leading to vasodilation [86]. -> atropine can block this reflex activation of vagus nerve, then it should lead to vasoconstriction. Actually, in Ref 85, it demonstrated that atropine acts as a competitive antagonist of norepinephrine and this action underlies its hypotensive effect.

AA: We thank the Reviewer 2 for his/her comment and have modified this paragraph according to his/her suggestion.

Rev2Q11: *L428: the atropine metabolism could also interfere with the oxygen blood saturation -> Is it demonstrated in Ref 89?

AA: We thank the Reviewer 2 for pointing out this inaccuracy and have modified both the sentence and the reference to avoid misleading discussion.

Rev2Q12: L432, L453: Ca2+ channel

AA: We thank the Reviewer 2 for pointing out this inaccuracy and have corrected the sentence.

Rev2Q13: *L485: Despite the cardiac action, reduction of pulse distension induced by JWH-018 did not enhance? after propranolol (possible a but not b adrenergic effect)

AA: We thank the Reviewer 2 for this comment and we have modified the sentence in the effort to clarify the point.

Rev2Q14: L512: 25-30 gr? L547: 4 µl/gr -> gram

AA: We thank the Reviewer 2 for pointing out this inaccuracy and have corrected the sentence.

Rev2Q15: L514: 80 cm2

AA: We thank the Reviewer 2 for pointing out this inaccuracy and have corrected the sentence.

Rev2Q16: *L643-958: Abbreviations: It is better to list in alphabetic order

AA: We thank the Reviewer 2 for his/her comment and have listed the abbreviations in alphabetic order.

Rev2Q17: *L659-: References (check all): inconsistent writing format for

(1) title: capital letter on the first word only (ref 6) vs. all words (ref 25)

(2) page number: ref 7: 1251-1261 vs. ref 11: 234-43

AA: We thank the Reviewer 2 for his/her comment and have corrected reference list in order to avoid inconsistent writing.
